# Uncovering and tailoring hidden Rashba spin–orbit splitting in centrosymmetric crystals

Linding Yuan [1,2], Qihang Liu[3,4], Xiuwen Zhang[5], Jun-Wei Luo[1,2,6], Shu-Shen Li[1,2,6] & Alex Zunger[3]

Hidden Rashba and Dresselhaus spin splittings in centrosymmetric crystals with subunits/sectors having non-centrosymmetric symmetries (the R-2 and D-2 effects) have been predicted theoretically and then observed experimentally, but the microscopic mechanism remains unclear. Here we demonstrate that the spin splitting in the R-2 effect is enforced by specific symmetries, such as non-symmorphic symmetry in the present example, which ensures that the pertinent spin wavefunctions segregate spatially on just one of the two inversion-partner sectors and thus avoid compensation. We further show that the effective Hamiltonian for the conventional Rashba (R-1) effect is also applicable for the R-2 effect, but applying a symmetry-breaking electric field to a R-2 compound produces a different spin-splitting pattern than applying a field to a trivial, non-R-2, centrosymmetric compound. This finding establishes a common fundamental source for the R-1 effect and the R-2 effect, both originating from local sector symmetries rather than from the global crystal symmetry per se.

[1] State Key Laboratory for Superlattices and Microstructures, Institute of Semiconductors, Chinese Academy of Sciences, Beijing 100083, China. [2] Center of Materials Science and Optoelectronics Engineering, University of Chinese Academy of Sciences, Beijing 100049, China. [3] Renewable and Sustainable Energy Institute, University of Colorado, Boulder, CO 80309, USA. [4] Institute for Quantum Science and Engineering, and Department of Physics, Southern University of Science and Technology, Shenzhen 518055, China. [5] College of Physics and Optoelectronic Engineering, Shenzhen University, Guangdong 518060, China. [6] Beijing Academy of Quantum Information Sciences, Beijing 100193, China. These authors contributed equally: Linding Yuan, Qihang Liu. Correspondence and requests for materials should be addressed to J.-W.L. (email: jwluo@semi.ac.cn) or to A.Z. (email: alex.zunger@colorado.edu)

Numerous physical effects and the technologies enabled by them are conditional on the presence of certain symmetries in the material that hosts such effects. Examples include effects predicated on the absence of inversion symmetry (non-centrosymmetric systems) such as the Dresselhaus effect[1], the Rashba effect[2], optical activity in non-chiral molecules[3], valley polarization and its derivative effects[4], and valley Hall effect in two-dimensional (2D) layered structures[5]. Although centrosymmetric systems are supposed to lack these effects, there is a large class of systems whose global crystal symmetry (GCS) is indeed centrosymmetric, but they consist of individual sectors with non-centrosymmetric local sector symmetry (LSS) (non-centrosymmetric site point groups). The term "hidden effect" refers to the general conditions where the said effect does exist even when the nominal GCS would disallow it. For example, the hidden Dresselhaus effect[6] occurs in the diamond-type structure of Silicon, where each atom has a non-centrosymmetric LSS (the tetrahedral $T_d$ point group) but the crystal as a whole has a centrosymmetric GCS (the octahedral $O_h$ group). The theoretical prediction[6] and subsequent experimental observations[7–13] of "hidden spin polarization" in non-magnetic centrosymmetric crystals triggered research on broader physical effects nominally disallowed under high GCS of systems, such as optical activity[14], intrinsic circular polarization[15], current-induced spin polarization[16,17], superconductor[18], piezoelectric polarization[6], and orbital polarization[19] in various centrosymmetric systems, as summarized in Table 1.

We use the designation "1" for cases where global inversion symmetry is absent (thus exhibiting the physical effects conditional on the absence of global inversion symmetry), as is the case of the conventional Rashba effect (R-1) or Dresselhaus effect (D-1). In parallel, we use the designation "2" for cases where the presence of global inversion symmetry hides the physical effects (conditional on the absence of symmetry), which is but revealed theoretically[6] and observed experimentally[7–13]. The latter is the case for the hidden Rashba effect (R-2) or hidden Dresselhaus effect (D-2)[6]. It is noteworthy that in R-2 or D-2 non-magnetic materials, even though the local spin polarization is nonzero, the net spin polarization remains zero (spin degeneracy), as imposed by the global inversion symmetry.

In the following, we build on our previous work ref. [6], the idea of hidden spin polarization and the general conditions for its existence—global inversion symmetry and existence of inversion-partner sectors with polar site point group symmetries—were introduced. Here we focus on the microscopic mechanisms at play and how can they be translated into design principles for selecting high-quality R-2 materials for future experiments. We (i) show a common denominator for both R-1 and R-2 Rashba splitting, i.e., both effects originate from the symmetries of the local inversion-partner sectors rather than the global symmetries of the systems. (ii) As net polarization requires then that the doubly degenerate states on the different sectors will be prevented from mixing, we point out the mechanism of symmetry-enforced wavefunction segregation, which prevents the doubly degenerate states on the different sectors from mixing. This is illustrated for the prototype compound in BaNiS$_2$ where the requisite symmetry is non-symmorphic operation. (iii) To clarify the difference between an R-2 compound and a trivial centrosymmetric compound, we investigate the evolution of the R-1 spin splitting from a R-2 spin splitting ("R-1 from R-2") by placing a tiny electric field on R-2, which breaks the global inversion symmetry. We find that even for a tiny applied field the ensuing $\alpha_R$ for "R-1 from R-2" far exceeds the effect in the "R-1 from trivial" case, highlighting that the observed R-2 spin splitting is not due to inadvertent breaking of the inversion symmetry in an ordinary centrosymmetric compound as recently thought[20]. This shows that angle resolved photoemission spectroscopy (ARPES) experiments can indeed probe band splitting genuinely coming from the hidden spin polarization and spin–orbit coupling (SOC), even if they are affected by surface sensitivity. This resolves another criticism raised by ref. [20] about potential difficulties in hidden spin polarization detection, namely the attribution of spin splitting to surface effects rather than to the bulk. This work sheds light on the view of the recent debate around the physical meaning and relevance of the "hidden spin polarization" concept and for the strong experimental and theoretical activity around it, motivated by the possibility to device materials with remarkable spin textures and technologically relevant properties. This finding offers clear experimental and computational frameworks to understand, tailor and use the R-2/D-2 effects.

## Results

### The evolution of R-2 into R-1 under an inversion symmetry-breaking electric field.

One might naively think that the observed R-2 spin splitting is due to inadvertent breaking of the inversion symmetry in an ordinary centrosymmetric compound.[21] Indeed, a centrosymmetric R-2 compound is distinct from a trivial centrosymmetric compound in that the former consist of individual polar sectors with non-centrosymmetric LSS (specifically, polar site point groups C$_1$, C$_2$, C$_3$, C$_4$, C$_6$, C$_{1v}$, C$_{2v}$, C$_{3v}$, C$_{4v}$ and C$_{6v}$). A tiny electric field applied to a centrosymmetric trivial material such as cubic perovskites[21] gives rise to a proportionally tiny spin splitting whose magnitude is proportional to the field. To clarify the difference between an R-2 compound and a trivial

---

**Table 1 Examples of reported hidden effects in centrosymmetric crystals**

| Polarization | Hidden functionality | Symmetry: LSS | Symmetry: GCS | Example |
|---|---|---|---|---|
| Spin | Dresselhaus effect | Non-CS and non-polar | CS | Si$_2$[6], Ge$_2$[6] |
| | Rashba effect | Polar | CS | BaNiS$_2$[10,38], LaOBiS$_2$[11,39,40] |
| | Spin–orbit torque in AFM | Non-CS | CS | CuMnAs[16], Mn$_2$Au[17] |
| Orbital | Atomic orbital | Non-CS | CS | Ge$_2$, GaAs[19] |
| Optical | Optical activity | Chiral | Non-chiral | [Cu(H$_2$O)(bpy)$_2$]$_2$[HfF$_6$]$_2$•3H$_2$O[13] |
| Valley | Circular polarization | Non-CS | CS | Bilayer TMDs[15] |
| Electric | Antipiezoelectric | Non-CS and non-polar exclude O | CS | BN[6], NaCaBi[6] |
| | Antipiezo- and antipyroelectric | Polar | CS | CdI$_2$[6], Bi$_2$Se$_3$[6] |
| SHG | IA-SHG-2 | Non-polar | CS | Si$_2$[6], NaCaBi[6] |
| | IA-SHG-2 and dp-SHG-2 | Polar | CS | MoS$_2$[6], Bi$_2$Se$_3$[6] |

Hidden effects are usually forbidden to exist in high global crystal symmetry (GCS) but are allowed in individual local sectors with low local sector symmetry (LSS). *AFE* antiferroelectricity, *CS* centrosymmetric, *Non-CS* non-centrosymmetric, *SHG* second harmonic oscillation, *IA-SHG-2* and *dp-SHG-2* denote hidden SHG effects as the site inversion asymmetry (*IA*) and site dipole field (*dp*) contained in local sectors induce the local SHG effects, which are compensated in global by opposite SHG effects from its inversion-partner sector. Non-CS polar point groups of LSS are explicitly C$_1$, C$_2$, C$_3$, C$_4$, C$_6$, C$_{1v}$, C$_{2v}$, C$_{3v}$, C$_{4v}$ and C$_{6v}$. Non-CS non-polar point groups of LSS are D$_2$, D$_3$, D$_4$, D$_7$, S$_4$, D$_{2d}$, C$_{3h}$, D$_{3h}$, T, T$_d$ and O

centrosymmetric compound, which is often confused[20], we investigate the evolution of the R-1 spin splitting from a R-2 spin splitting ("R-1 from R-2") by using the first-principles calculations on R-2 compounds and placing on it a tiny electric field that breaks the global inversion symmetry.

An example of R-2 compounds is BaNiS$_2$[10], which is a five-coordinated Ni(II) structure consisting of puckered 2D layers of edge-sharing square pyramidal polyhedral and crystalizes in the tetragonal system, space group P4/nmm. Conductivity and susceptibility measurements[22,23] indicate that it is a metallic Pauli Paramagnet. Our DFT + U calculation ($U = 3$ eV, $J = 0.95$ eV) also predicts a low-temperature anti-ferromagnetic phase with local Ni moments of $\pm 0.7\,\mu_B$ for bulk ($\pm 0.6\,\mu_B$ for a monolayer) where the anti-ferromagnetic phase is slightly more stable than the non-magnetic model by just 43 meV(f.u)$^{-1}$ for bulk and 28 meV(f.u)$^{-1}$ for monolayer. These DFT + U calculations had reported that BaNiS$_2$ undergoes a phase transition from paramagnetic to anti-ferromagnetic as increasing the used $U$-value from 2 to 3 eV. Given the difficulty

of estimating the proper $U$-value in the $+\,U$ framework and experimental (conductivity and susceptibility) observation[22,23] of metallic Pauli Paramagnet, in this work we nevertheless adopt a non-magnetic phase for BaNiS$_2$ to avoid the unnecessary complications from magnetic orders. Our relaxed lattice constants and interatomic distances in the non-magnetic General Gradient Approximation (GGA) calculation agrees with the measured result within ~1%[10,22]. In the non-magnetic model, BaNiS$_2$ possesses both inversion symmetry and time-reversal symmetry; in the presence of SOC, each energy band is even-fold degenerate and thus has no R-1 spin splitting.

Figure 1a shows the structure of a monolayer of this centrosymmetric crystal, which has two separated crystallographic sectors–$S_\alpha$ and its inversion partner $S_\beta$ (shown in Fig. 1a as red and blue planes, respectively); each sector contains a single B atom (here, $B =$ Ni, Pd, or Pt) with a polar site group $C_{4v}$, having its local internal dipole field[10] (calculated and shown below). We focus our attention on the lowest four conduction bands (including spin) around the $\bar{X}$ point

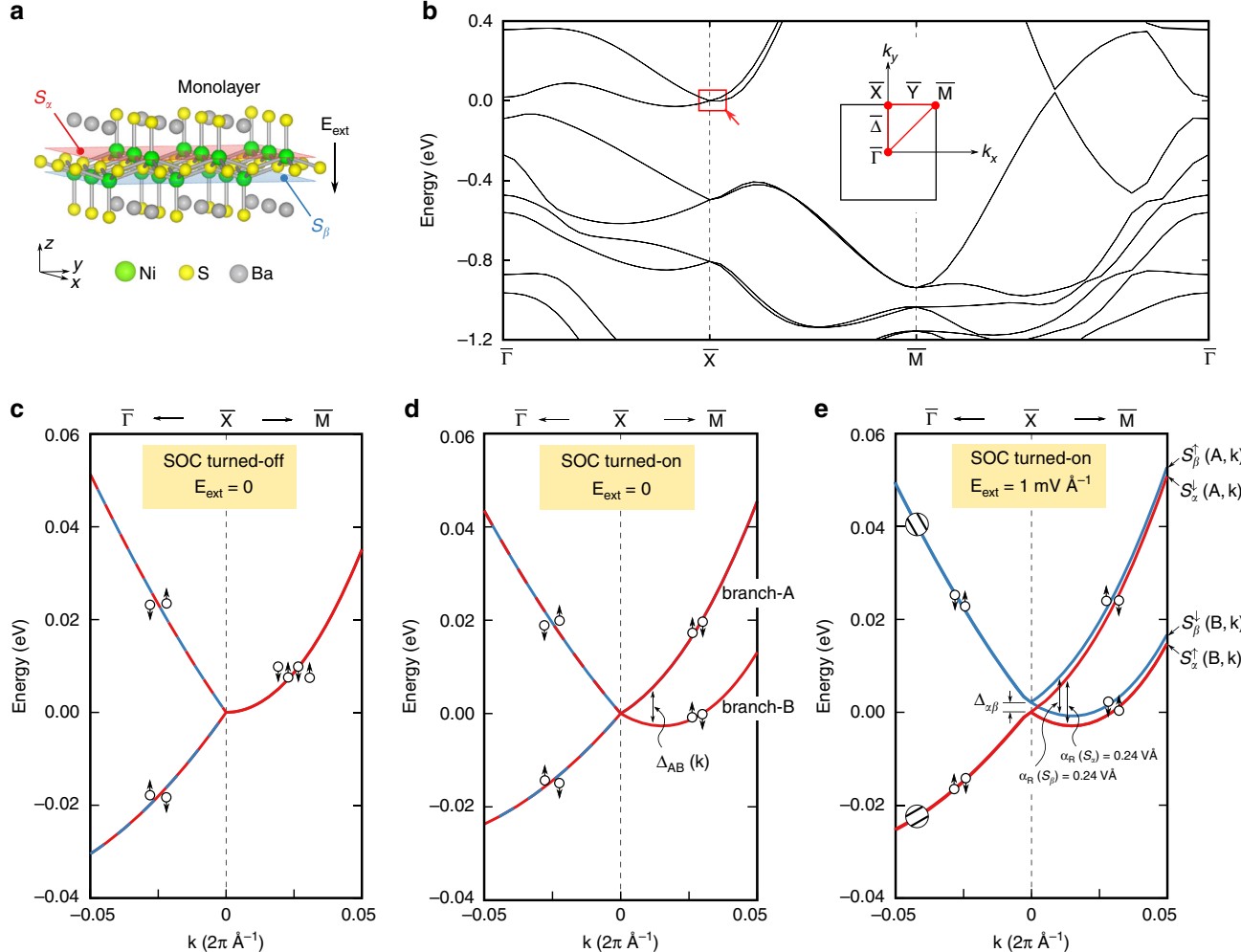

**Fig. 1** The crystal structure and energy bands of the monolayer BaNiS$_2$. **a** The crystal structure of a centrosymmetric monolayer of BaNiS$_2$ taken from the bulk with P4/nmm space group, showing its two inversion-partner sectors $S_\alpha$ and $S_\beta$. **b** Energy band dispersion of the monolayer in an extended zone. The Rashba bands of interest are highlighted in red square. Insert shows schematically the 2D Brillouin zone of the monolayer. **c–e** Zoom-in the energy dispersion of the lowest four conduction bands near the $X$ point along $\bar{X} - \bar{\Gamma}$ and $\bar{X} - \bar{M}$ directions when SOC is turned off (**c**) and turned on (**d**, **e**). Relative to the result shown in **d**, in case shown in **e** we apply a small electric field of 1 mV Å$^{-1}$ to the monolayer along the z-direction, as schematic digram shown in **a**, to break the inversion symmetry. The inversion symmetry-breaking electric field lifts the degeneracy of both branches A and B into the $S_\alpha$-Rashba band and the $S_\beta$-Rashba band, with an energy separation at the $X$ point denote as $\Delta_{\alpha\beta}$. The band with its wavefunction segregated on the sector $S_\alpha$ is represented by red and on the on sector $S_\beta$ by blue. Arrows are used to illustrate the spin orientation

(highlighted with a red square in Fig. 1b). Figure 1c shows that when SOC is turned off in the first-principles calculations, one finds along high-symmetry path $\bar{X} - \bar{M}$ a single, fourfold degenerate band whose degeneracy is imposed by the non-symmorphic screw-axis symmetry $\{C_{2x}|(a/2, 0, 0)\}$; $\{C_{2y}|(0, a/2, 0)\}$ (explained in Supplementary Note 2 and 3). When SOC is turned on, the fourfold degenerate band splits into two branches A and B (Fig. 1d) and each branch is doubly degenerate and has two orthogonal spin components. The applied out-of-plane electric field external electric field generates asymmetric potential on the two inversion-partner sectors and thus breaks the global inversion symmetry, but conserves the time-reversal symmetry.

The spin degeneracy of both branches A and B along $\bar{X} - \bar{M}$ and at the $\bar{X}$ point is lifted upon application of an external field $\mathbf{E}_{ext}$, as shown in Fig. 1e. This splitting, denoted $\Delta_{\alpha\beta}$, occurs at the time-reversal invariant (TRI) $\bar{X}$ point and is dependent linearly on $\mathbf{E}_{ext}$ (see below). The finite splitting at the TRI point rules out the Rashba effect as the origin of the splitting of the two spin components of branch A (and branch B) along $\bar{X} - \bar{M}$. Figure 2a indeed shows that the spin-down component of the high-energy branch A and the spin-up component of the low-energy branch B have wavefunctions confined in sector $S_\alpha$, and thus pair as one orbital band (hereafter, termed $S_\alpha$-Rashba band). The spin-up component of the branch A and the spin-down component of the branch B possess wavefunctions confined in

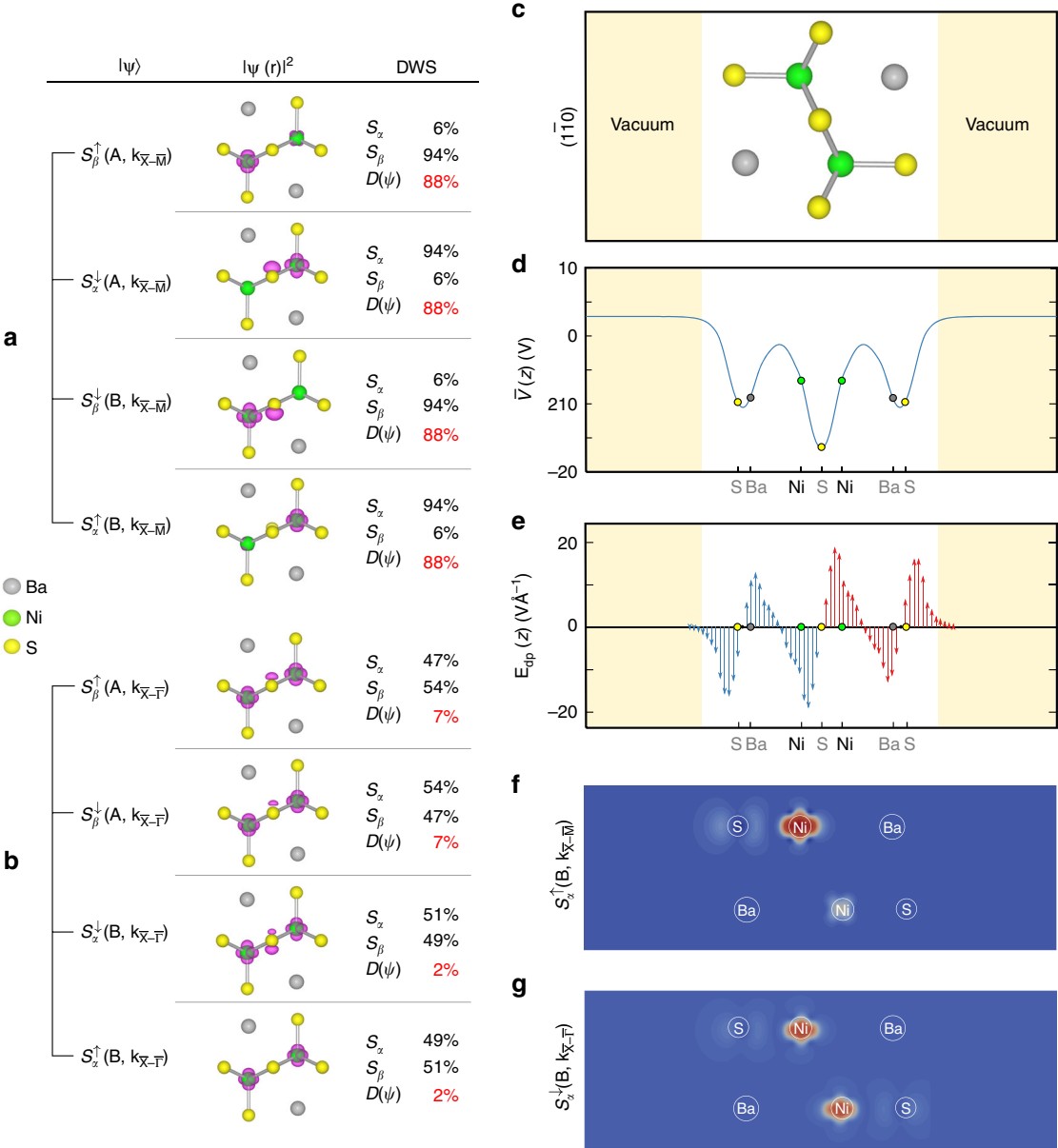

**Fig. 2** Wavefunction segregation and local dipole fields in BaNiS$_2$ monolayer. **a, b** Charge density of the lowest four conduction bands at $\mathbf{k}_{\bar{X}-\bar{\Gamma}} = (0, 0.475, 0)(2\pi/a)$ and $\mathbf{k}_{\bar{X}-\bar{M}} = (0.025, 0.5, 0)(2\pi/a)$, respectively. The isosurface of charge density is represented by purple. The Ni, S, and Ba atoms are represented by green, yellow, and gray balls, respectively. The degree of wavefunction segregation and the percentage of the charge density localized on the sectors $S_\alpha$ and $S_\beta$ are also listed for each state. **c** The crystal structure of the monolayer BaNiS$_2$, a view perpendicular to the $(1\bar{1}0)$ plane. **d** Planar-averaged crystal potential of the monolayer BaNiS$_2$. **e** The $z$-component of the internal local dipole fields $\mathbf{E}_{dp}(z) = (1/e)\partial\bar{V}(z)/\partial z$ along the $z$-direction. Red arrows indicate the dipole fields within the sector $S_\alpha$ and blue arrows for the dipole fields within the sector $S_\beta$. **f, g** Charge density of the $S_\alpha^\uparrow(B, \mathbf{k}_{\bar{X}-\bar{\Gamma}})$ and $S_\alpha^\uparrow(B, \mathbf{k}_{\bar{X}-\bar{M}})$ states of the monolayer BaNiS$_2$ in the absence of external fields

sector $S_\beta$ (hereafter, termed $S_\beta$-Rashba band). We therefore identify the splitting $\delta E_{AB}(\mathbf{k})$ as a consequence of the R-2 effect quantified by a Rashba parameter $\alpha_R(R2) = 0.24$ V·Å. The applied electric field further adds/subtracts the R-1 spin splitting to/from the R-2 splitting $\delta E_{AB}(\mathbf{k})$ of the $S_\alpha$- and $S_\beta$-Rashba bands, respectively, along the $\bar{X} - \bar{M}$ direction. Figure 3a shows the corresponding Rashba parameters $\alpha_R = \delta E_{AB}(\mathbf{k} - \bar{X})/2(\mathbf{k} - \bar{X})$, which exhibits a linear response to $\mathbf{E}_{ext}$: $\alpha_R$ of the $S_\alpha$-Rashba band increases and the $S_\beta$-Rashba band decreases at rates of the same magnitude but opposite sign as increasing $\mathbf{E}_{ext}$. The extrapolations of these two $\alpha_R$ functions cross at $\mathbf{E}_{ext} = 0$, giving rise to $\alpha_R = 0.24$ V·Å, a value being the same as the (zero field) R-2 spin splitting $\alpha_R(R-2)$.

The magnitude of the R-2 spin splitting can be determined unambiguously by placing on a candidate R-2 compound an electric field, then extrapolating to the zero field to uncover a finite, zero-field (R-2) Rashba parameter. The significant magnitude illustrated above of the ensuing $\alpha_R$ for "R-1 from R-2" relative to the "R-1 from trivial" scenario highlights the fact that the R-1 spin splitting is inherited from the R-2 effect in bulk Rashba systems, i.e., from the local asymmetric dipole fields of the individual sectors. This finding obviates the concern of Li and Appelbaum[20] who suggested that the Rashba surface spin splitting detected experimentally (e.g., via ARPES) might originate from the unavoidable inversion symmetry-broken surface, as this contribution is indistinguishable from bulk R-2 effect.

**Avoided compensation of the R-2 spin polarization in BaNiS$_2$ enforced by non-symmorphic symmetry.** We next clarify under what circumstances the hidden R-2 effect can be large or small. This physics can be gleaned by looking at a single non-magnetic centrosymmetric R-2 ABX$_2$ system in two different directions in the Brillouin zone (BZ). Figure 1 shows that these R-2 bands along $\bar{X} - \bar{M}$ and $\bar{X} - \bar{\Gamma}$ directions exhibit two different types of spin-splitting behaviors associated with the distinct transformation properties of the wavefunction under non-symmorphic glide reflection symmetry (see Supplementary Note 3 for details). This realization then would help us establish the distinguishing features of R-1 vs. R-2 materials.

**Wavefunction segregation causes sizable R-2 spin splitting along $\bar{X} - \bar{M}$ direction.** To quantify the degree of wavefunction segregation (DWS) of the wavefunction, we introduce a measure $D(\varphi_{\mathbf{k}})$ for states $\varphi_{\mathbf{k}}$ at the wavevector $\mathbf{k}$, where

$$D(\varphi_{\pmb{k}}) = \left| \frac{P_{\varphi_{\mathbf{k}}}(S_\alpha) - P_{\varphi_{\mathbf{k}}}\left(S_\beta\right)}{P_{\varphi_{\mathbf{k}}}(S_\alpha) + P_{\varphi_{\mathbf{k}}}\left(S_\beta\right)} \right|, \quad (1)$$

and

$$P_{\varphi_{\mathbf{k}}}(S_{\alpha,\beta}) = \int_{\Omega \in S_{\alpha,\beta}} |\varphi_{\mathbf{k}}(\mathbf{r})|^2 d^3\mathbf{r}. \quad (2)$$

$P_{\varphi_{\mathbf{k}}}(S_\alpha)$ is the component of the wavefunction $\varphi_{\mathbf{k}}$ localized on the sector $S_\alpha$. The DWS explicitly quantifies the locality of wavefunction, in contrast to the implicit measure[10] by means of the integral of the local spin density operator restricted on a given sector.

It is evident that $D(\varphi_{\mathbf{k}}) = 0$ for a wholly delocalized wavefunction over two inversion-partner sectors, whereas, $D(\varphi_{\mathbf{k}}) = 100\%$ indicates that the wavefunction is entirely confined either on sector $S_\alpha$ or sector $S_\beta$. One expects, in general, that any linear combination of two degenerate states should still be an eigenstate

and prevent us from obtaining a unique DWS for the energy-degenerate bands.[20] However, we demonstrated in Supplementary Note 3 that, in R-2 compounds, the symmetry of the wavevectors along $\bar{X} - \bar{M}$ direction prohibits the mixing of two degenerate states arising from two inversion-partner sectors ($S_\alpha$ and $S_\beta$), respectively, as a result of the glide reflection symmetry, and hence dissociates any linear combinations of the degenerate states for tracing back to the symmetry-enforced segregated states. Santos-Cottin et al.[10] had shown the localization of wavefunction in BaNiS$_2$ to provide the basis to decouple two effective Rashba Hamiltonians associated with each sector. Our calculations also (Fig. 2a) show segregated wavefunctions (localized either on sector $S_\alpha$ or $S_\beta$) and $D(\varphi_{\mathbf{k}}) = 88\%$ ($\mathbf{k} = (0.025, 0.5, 0)(2\pi/a)$, here $a$ is the lattice constant, for both spin components of doubly degenerate branches A and B along $\bar{X} - \bar{M}$ direction. This fact obviates the concern of validity of hidden spin-splitting theory due to the possible lack of gauge invariance, raised by Li and Appelbaum[20].

The relation between wavefunction segregation and the R-2 effect can be appreciated as follows: in 2D quantum wells or heterojunctions, one obtains the Rashba parameter $\alpha_R$ due to the R-1 effect as[24]

$$\alpha_{R,i} = \left\langle r_{R,i} \cdot \mathbf{E}(\mathbf{r}) \right\rangle \quad (3)$$

where $r_{R,i}$ is a material-specific Rashba coefficient of the $i$th-band, the electric field $\mathbf{E}(\mathbf{r}) = (1/e)\nabla V$ is the local gradient of the crystal potential $V$, and angular brackets indicate an average of the local Rashba parameter $r_{R,i}\mathbf{E}(\mathbf{r})$ of the well and barrier materials weighted by the wavefunction amplitude. In a crystal without external fields, the electric field originates from the local dipole and is termed $\mathbf{E}_{dp}(\mathbf{r})$, which does not have to vanish at all atomic sites even in centrosymmetric systems. Figure 2e shows the $x$–$y$ planar-averaged internal local dipole fields $\mathbf{E}_{dp}(z)$ in the monolayer BaNiS$_2$. It exhibits that $\mathbf{E}_{dp}(z)$ varies rapidly within a single sector and is inversion through a point located on the sulfur atom (or point reflection). The internal dipole fields are finite (and in fact atomically large) within a single sector, whereas the sum over both inversion-partner sectors is zero as expected. The segregation of wavefunctions on a single sector with $D(\varphi_{\mathbf{k}}) = 88\%$ for states along $\bar{X} - \bar{M}$ direction indicates that this band experiences a net effective field of the internal dipole fields within a single sector (as illustrated in Fig. 2f) and is immune to full compensation from the opposite dipole fields within its inversion-partner sector. According to Eq. (3), a finite Rashba parameter $\alpha_R$ is thus obtained for R-2 bands along $\bar{X} - \bar{M}$ direction. Thus, the large R-2 effect along this BZ direction originates from wavefunction segregation on each of the two inversion-partner sectors, avoiding mutual compensation of local dipolar electric fields.

**Wavefunction delocalization leading to vanishing R-2 spin splitting along the $\bar{X} - \bar{\Gamma}$ direction.** In sharp contrast to the $\bar{X} - \bar{M}$ direction, Fig. 1c shows that along $\bar{X} - \bar{\Gamma}$ direction these four bands already split into two doublets even in the absence of SOC and the magnitude of their splitting is barely changed after turning on the SOC. We attribute such band splitting to symmetry allowed interaction between states stemming from two inversion-partner sectors $S_\alpha$ and $S_\beta$ (see Supplementary Note 3). Thereby, we denote two spin components of the branch A by $S_{\alpha/\beta}^\downarrow(A, \mathbf{k}_{\bar{X}-\bar{\Gamma}})$ and $S_{\alpha/\beta}^\uparrow(A, \mathbf{k}_{\bar{X}-\bar{\Gamma}})$, respectively, whereas, for branch B we use $S_{\alpha/\beta}^\downarrow(B, \mathbf{k}_{\bar{X}-\bar{\Gamma}})$ and $S_{\alpha/\beta}^\uparrow(B, \mathbf{k}_{\bar{X}-\bar{\Gamma}})$. The wavefunction of the spin-down component of the branch A is 49% confined, and that of branch B is 51% confined in sector $S_\alpha$, respectively, so as

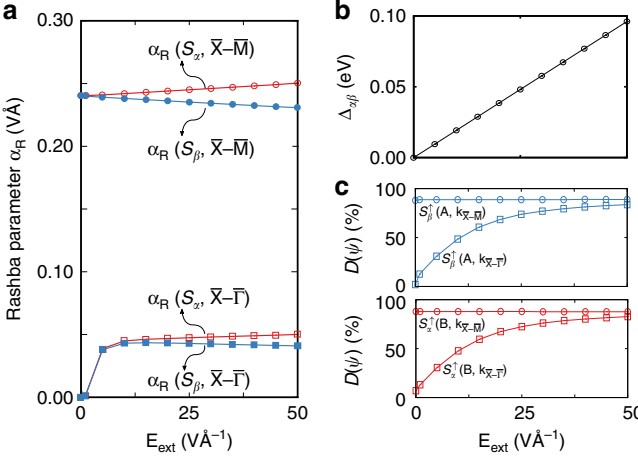

**Fig. 3** The evolution of Rashba physics under electric field in monolayer BaNiS₂. **a** The Rashba parameters of the spin-splitting bands segregated on the sector $S_\alpha$ (red empty squares or circles) and sector $S_\beta$ (blue solid squares or circles), respectively, along $\bar{X} - \bar{\Gamma}$ (square) and $\bar{X} - \bar{M}$ (circle) directions as a function of applied electric field. **b** Electric field induced energy separation ($\Delta_{\alpha\beta}$) between the $S_\alpha$-Rashba band and the $S_\beta$-Rashba band at $X$ point. **c** Degree of wavefunction segregation of branch A (upper panel) and branch B (lower panel) along $\bar{X} - \bar{\Gamma}$ and $\bar{X} - \bar{M}$ directions, respectively, as functions of the applied electric field. It is noteworthy that the Rashba parameter shown in **a** is a sum of dipole fields weighted by corresponding wavefunction amplitudes and are not necessary to display a simple linear correlation with the DWS (shown in **c**), a degree of wavefunction segregation defined in Eqs. (1) and (2)

Fig. 2b shows DWS is $D\left(S^\downarrow_{\alpha/\beta}(A, \mathbf{k}_{\bar{X}-\bar{\Gamma}})\right) = D\left(S^\downarrow_{\alpha/\beta}(B, \mathbf{k}_{\bar{X}-\bar{\Gamma}})\right) = 2\%$ for spin-down components of both A and B branches. Similarly, the wavefunction of the spin-up component of the branch A is 43% confined, and that of branch B is 57% confined in sector $S_\alpha$ so DWS $D\left(S^\uparrow_{\alpha/\beta}(A, \mathbf{k}_{\bar{X}-\bar{\Gamma}})\right) = D\left(S^\uparrow_{\alpha/\beta}(B, \mathbf{k}_{\bar{X}-\bar{\Gamma}})\right)$ is 14% for spin-up components. Thus, the wavefunctions of the $\bar{X} - \bar{\Gamma}$ bands are essentially delocalized over both inversion-partner sectors $S_\alpha$ and $S_\beta$. Such wavefunction delocalization naturally leads to a complete compensation of the undergoing local internal dipole fields within $S_\alpha$ by that within $S_\beta$, when each local dipole weighted by its wavefunction amplitudes gives rise to zero average Rashba parameter $\alpha_R$ according to Eq. (3).

**Unification of R-1 and R-2 into a single theoretical framework.** The smooth "R-1 from R-2" evolution (Fig. 3a) suggests that when applying an external electric field $\mathbf{E}_{ext}$ to an R-2 system, the electric field $\mathbf{E}(\mathbf{r})$ acting on electrons is a superposition of $\mathbf{E}_{ext}$ and the internal local dipole (dp) electric fields $\mathbf{E}_{dp}(\mathbf{r})$,

$$\mathbf{E}(\mathbf{r}) = \mathbf{E}_{dp}(\mathbf{r}) + \mathbf{E}_{ext} \qquad (4)$$

Thus, both R-1 and R-2 spin splitting have a common fundamental source being the dipole electric fields of the local sectors rather than from the global crystal asymmetry per se. Such local dipole electric field "lives" within individual local sectors. The fundamental difference between R-1 and R-2 effects is that in R-2 the spin splitting is hidden by the overlapping energy bands arising from two inversion-partner sectors, whereas in the R-1 case such overlap is forbidden by the global inversion asymmetry.

Figure 1e also shows that the applied electric field lifts the spin degeneracy of the bands along $\bar{X} - \bar{\Gamma}$ direction and raises $\alpha_R$ linearly from zero at $\mathbf{E}_{ext} = 0$ to saturation at $|\mathbf{E}_{ext}| = 10$ mV Å⁻¹ at an odd large rate. This behavior is in striking contrast to the

linear field dependence of the bands along $\bar{X} - \bar{M}$ direction (see Fig. 3a). Such unusual field dependence of $\alpha_R$ confirms again that R-2 spin splitting evolves smoothly to the R-1 spin splitting upon the breaking of the global inversion symmetry, regarding the bands along $\bar{X} - \bar{\Gamma}$ direction have vanishing R-2 spin splitting with $\alpha_R(R2) = 0$ in the absence of an external field. Upon application of electric field, the delocalized wavefunctions of the $\bar{X} - \bar{\Gamma}$ bands become gradually segregated on one of two inversion-partner sectors as a result of Stark effect[25]. Subsequently, Fig. 3c shows that the applied field amplifies substantially the DWS (Eq. (1)) of the spin-up component of both branches from 14% to > 80% as the magnitude of $\mathbf{E}_{ext}$ increases from 0 to 50 mV Å⁻¹. However, $D(\varphi_\mathbf{k})$ is barely changed once $\mathbf{E}_{ext} > 50$ mV Å⁻¹ (saturation field). It is noteworthy that DWS of the corresponding spin-down components is not shown but has a similar response to the applied electric field. It is straightforward to learn that the internal electric dipole fields acting on these bands become uncompensated as their wavefunctions change into segregation on a single sector, evoking the R-2 effect with its strength highly related to $D(\varphi_k)$ according to Eq. (3). The rapid amplification of $D(\varphi_k)$ by the applied electric field explains that the (unusual) rapid rise of $\alpha_R$ for those bands along $\bar{X} - \bar{\Gamma}$ direction is mainly due to the enhancement of the wavefunction segregation rather than to the increase of the total electric dipole field.

When $|\mathbf{E}_{ext}|$ reaches ~25 mV Å⁻¹, $\alpha_R$ of both high- and low-energy doublets become linear field-dependent but in rates of opposite signs, which is in a similar field dependence as that along $\bar{X} - \bar{M}$ direction. Figure 3c shows that the response of $D(\varphi_k)$ of the $\bar{X} - \bar{M}$ bands to $\mathbf{E}_{ext}$ is, however, barely modified by the external field, indicating those states remain fully localized on one of two inversion-partner sectors. The linear change of $\alpha_R$ along $\bar{X} - \bar{M}$ direction as shown in Fig. 3a thus arises entirely from the external field induced asymmetry, i.e., in Eq. (3) the change $\alpha_R$ is solely arising from the electric field. The calculated Rashba parameter of the R-2 spin splitting can be explained regarding the model of the R-1 spin splitting (Eq. (3)), indicating a unified theoretical view for both R-1 and R-2 effects in bulk systems. Specifically, the effective electric field that promotes either R-1 and/or R-2 Rashba effects is a superposition of the applied external electric field plus the internal local electric fields originating from the dipoles of the individual local sectors, weighted by the wavefunction amplitude on the corresponding sectors.

We also apply this unifying theoretical framework to a non-layered R-1 example, the α-SnTe[6] or similarly the α-GeTe (a standard ferroelectric bulk R-1 compound predicted in 2013[26] and experimentally confirmed in 2016[27,28]), where one can identify two inversion-partner sectors and the corresponding wavefunction becomes segregated due to the lack of inversion symmetry in the rhombohedral phase (details see Supplementary Note 1). According to the unified model described by Eq. (3), such wavefunction segregation gives a residual dipole field felt by band states and thus give rise to a finite Rashba spin splitting, similar to that of R-2 spin splitting in BaNiS₂. As displacing the Te atom from Sn along [111] direction, the α-GeTe will undergo a phase transition from non-centrosymmetric rhombohedral phase to centrosymmetric rocksalt phase. We demonstrate that in the centrosymmetric rocksalt phase wavefunctions are evenly distributed among two inversion-partner sectors, leading to a perfect compensation of the local dipole fields and thus vanishing Rashba effect in the centrosymmetric rocksalt phase according to Eq. (3).

**Design principles for increasing the strength of the R-2 effect.** R-2 materials[6] are defined by having global inversion symmetry

and two recognizable inversion-partner sectors with polar site point group symmetries. Designing R-2 materials possessing large hidden spin splitting and hence strong local spin polarization can benefit from two additional design principles:

(i) Minimizing the mixing and entanglement of the wavefunction on the different inversion partners sectors. Here we point to a nontrivial mechanism of symmetry-enforced wavefunction segregation, keeping the doubly degenerate states on the different sectors from mixing (in contrast to the trivial physical separation of the two inversion-partner sectors). It is noteworthy that R-1 compounds do not have to maintain segregation-inducing symmetries to have Rashba effect, because its inversion asymmetry alone ensures the avoidance of wavefunction entanglement by lifting the degeneracy of states from the two partner sectors, as illustrated in Supplementary Note 1 for rhombohedral SnTe. The wavefunction segregation enforcing symmetry illustrated here is the non-symmorphic symmetry along the $\bar{X} - \bar{M}$ direction in the $BaNiS_2$ BZ. Other segregation enforcing symmetry operations may exist in general cases, but they have not been discovered yet.

(ii) Instilling strong local dipole fields, i.e., designing individual sectors with maximal asymmetry of the local potential within the sector. Thus, whereas the creation and enhancing Rashba effect in conventional (e.g., interfacial) Rashba materials[2,24] entails, by tradition, breaking inversion symmetry, here our design principles for Rashba effect in centrosymmetric compounds focuses on using other symmetry operations that enhance segregation and avoid mixing.

Applying the design principles (i) and (ii) one could design strong R-2 materials via selecting compounds where the wavefunctions are concentrated in real space locations that have a larger magnitude of local dipole fields. An example illustrated here is $BaNiS_2$. Such wavefunction segregation can be tailored through application of an external electric field, strain, atom mutation, or modifications of the polar cation ordering.[24] This is illustrated by the rapid rise of $\alpha_R$ vs. field for bands along $\bar{X} - \bar{\Gamma}$ direction (Fig. 3a), demonstrating tailoring of the R-2 effect. For instance, Otani and colleagues[29] have recently found a strong correlation between the charge density distribution and the strength of the Rashba effect at non-magnetic metal/$Bi_2O_3$ interfaces. Furthermore, the unexpected rapid rise of $\alpha_R$ vs. field for bands along $\bar{X} - \bar{\Gamma}$ direction (Fig. 3a) implies that one might effectively tune the strength of R-2 effect. We thus present an alternative mechanism for boosting the strength of the Rashba effect, which is commonly achieved by enhancing the breaking of inversion symmetry.

## Methods

**First-principles band structure calculation.** Electronic structures are calculated using density functional theory (DFT)[30–32]-based first-principles methods within the GGA[33] implemented in the Vienna Ab initio simulation package (VASP)[34]. A plane-wave expansion up to 400 eV is applied and a $\Gamma$-centered $16 \times 16 \times 1$ Monkhorst-Pack[35] **k**-mesh is used for the BZ sampling. The lattice constants used in the first-principles calculations are taken directly from the experimental data. The monolayer slab of $BaNiS_2$ are separated by a 17.8 Å vacuum layer. We adopt the GGA + U method[36] to account the on-site Coulomb interaction of localized Ni-3$d$ orbitals. We follow the approach proposed by Neugebauer and Scheffler[37] to apply a uniform electric field to monolayer $BaNiS_2$ slab in the calculations. This approach treats the artificial periodicity of the slab by adding a planar dipole sheet in the middle of the vacuum region. The strength of the dipole is calculated self-consistently such that the electrostatic field-induced dipole is compensated for. For the calculations including the spin–orbit interaction, the spin quantization axis set to the default (0 +, 0, 1) (the notation 0 + implies an infinitesimal small positive number in the x-direction) with zero atomic magnetic moments. The VASP

configuration files and related codes that support the findings of this study are available from the corresponding authors upon reasonable request.

## Data availability
The data that support the findings of this study are available from the corresponding authors upon reasonable request.

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

## Acknowledgements

J.L. and S.L. were supported by the National Natural Science Foundation of China (NSFC) under Grant Number 61888102. J.L. was also supported by the National Young 1000 Talents Plan. Work of A.Z. and Q.L at CU Boulder was supported by the National Science foundation NSF Grant NSF-DMR-CMMT Number DMR-1724791. Q.L. was also supported by the NSFC under Grant Number 11874195. X.Z. was supported by the NSFC under Grant Number 11774239.

## Author contributions

L.Y. performed the electronic structure calculations, prepared the figures, and developed the tight-binding models with the help of Q.L. J.L. proposed the research project. J.L. and A.Z. established the project direction and conducted the analysis, discussion, and writing of the paper with input from L.Y., Q.L., and X.Z. S.L. provided the project infrastructure and supervised L.Y.'s study.

## Additional information

**Competing interests:** The authors declare no competing interests.

