## [Peer Review File · Nature Communications]

Reviewers' comments:

Reviewer #1 (Remarks to the Author):

In the manuscript under consideration, the authors revise the theory of the hidden Rashba spin-polarization in centrosymmetric crystals previously published by the same group in [Nat. Phys. 10 387 (2014)]. By explicitly taking the case of BaNiS₂, a member of the R-2 family, the authors show that, by applying a small external electric field, a conventional R-1 Rashba phase can be obtained out of the non-conventional hidden Rashba R-2 phase. Based on accurate first-principles calculations, it is shown that the wavefunctions of different Rashba branches segregate spatially on the opposite inversion-partner sectors. Each sector has a local symmetry that produces a net dipole moment. The main two results of this work are: (i) the segregation of the wavefunction which leads to the R-2 type spin splitting crucially depends on the non-symmorphic symmetry of the crystal; (ii) a common denominator is found for both R-1 and R-2 Rashba splitting, i.e. both effects originate from the symmetries of the local inversion-partner sectors rather than the global symmetries of the systems.

Technically speaking, I don't have any doubts about the accuracy, reliability, reproducibility, and correctness of the results. My main concern is if or not the present manuscript represents a significant advance compared to what has already been published 4 years ago in the Nat. Phys. The segregation of the wavefunction was already proposed. I can cite here some text from the NatPhys: "A key observation here is that in the R-2 and D-2 effects, the spin polarization of the energetically degenerate bands is spatially segregated into a dominant spin texture for one real-space sector, whereas the opposite spin texture is associated with its inversion partner." Of course, I understand that now the authors show that the non-symmorphic symmetry is at the heart of the segregation process. But my question is: how general is this mechanism? I can imagine R-1 type Rashba systems that belong to space-groups with a symmorphic global crystal symmetry. Moreover, it seems that is a general result that R-1 and R-2 Rashba both originate from the local sector symmetries. But if I consider one of the simplest R-1 ferroelectrics, let's say alpha-Sn, how can I identify the two inversion-partner sectors such that the R-1 spin-splitting in alpha-Sn originates from the local symmetries of the sectors rather than being a consequence of the global inversion symmetry breaking? The study presented in this manuscript seems specific to layered systems.

Some very minor observations:

(i) In the caption of Fig. 1e, the electric field magnitude is reported to be 5 mV/Å. However in the text as well as in the panel e $E_{\text{ext}}=1$ mV/Å.

(ii) The measure for the degree of segregation is shortened as DOD. I imagine that the authors don't want to use DOS to avoid confusion with the density of states. What the second D stand for?

(iii) In the methods section, details on the first-principles calculations within an external electric field are missing.

Comments of Reviewer #1 and authors' reply:

1. Referee comment: *In the manuscript under consideration, the authors revise the theory of the hidden Rashba spin polarization in centrosymmetric crystals previously published by the same group in [Nat. Phys. 10 387 (2014)]. By explicitly taking the case of BaNiS₂, a member of the R-2 family, the authors show that, by applying a small external electric field, a conventional R-1 Rashba phase can be obtained out of the non-conventional hidden Rashba R-2 phase. Based on accurate first-principles calculations, it is shown that the wavefunctions of different Rashba branches segregate spatially on the opposite inversion-partner sectors. Each sector has a local symmetry that produces a net dipole moment.*

The main two results of this work are:

- (i) the segregation of the wavefunction which leads to the R-2 type spin splitting crucially depends on the non-symmorphic symmetry of the crystal;*
- (ii) a common denominator is found for both R-1 and R-2 Rashba splitting, i.e. both effects originate from the symmetries of the local inversion-partner sectors rather than the global symmetries of the systems.*

Technically speaking, I don't have any doubts about the accuracy, reliability, reproducibility, and correctness of the results.

Author reply: Thank you for your appreciation of our work.

2. Referee comment: *My main concern is if or not the present manuscript represents a significant advance compared to what has already been published 4 years ago in the Nat. Phys. The segregation of the wavefunction was already proposed. I can cite here some text from the Nat Phys: "A key observation here is that in the R-2 and D-2 effects, the spin polarization of the energetically degenerate bands is spatially segregated into a dominant spin texture for one real-space sector, whereas the opposite spin texture is associated with its inversion partner."*

Author reply: This concern about the level of novelty relative to the 2014 paper is well addressed by Referee #2:

"The manuscript unveils the importance of the wave function segregation to explain the intriguing physical properties of the R-2 materials. This major property is rationalized for the first time based on symmetry arguments that constrain the form of the Hamiltonian along high-symmetry directions in the Brillouin zone. Moreover, starting from these considerations, an attempt to unify R-1 and R-2 materials is made. The paper is original, well organized. More importantly, this work is timely, in view of the recent debate around the physical meaning and relevance of the "hidden spin polarization" concept, and for the strong experimental and theoretical activity around it, motivated by the possibility to devise materials with remarkable spin textures and technologically relevant properties."

To make the novelty of this work stands out much more clearly relative to our previous articles, we have now listed the main ideas at the end of the introduction (bottom of p. 2 continuing on p. 3). This serves to give the reader a synopsis of what's exciting:

“In our previous work Ref. 6, the idea of hidden spin polarization and the general conditions for its existence -- global inversion symmetry and existence of inversion-partner sectors with non-centrosymmetric site point group symmetries-- was introduced. In the present work, we focus on the microscopic mechanisms at play and how can they be translated into “design principles “for selecting high-quality R-2 materials for future experiments. We (i) show a common denominator for both R-1 and R-2 Rashba splitting, i.e. both effects originate from the symmetries of the local inversion-partner sectors rather than the global symmetries of the systems. (ii) Since net spin polarization requires that the doubly degenerate states on the different sectors be prevented from mixing, we point out the mechanism of symmetry-enforced wavefunction segregation, that prevents the doubly degenerate states on the different sectors from otherwise mixing. This is illustrated for the prototype compound in BaNiS_2 where the requisite symmetry is the non-symmorphic operation. (iii) To clarify the difference between an R-2 compound and a trivial centrosymmetric compound we investigate the evolution of the R-1 spin-splitting from a symmetry-broken R-2 spin-splitting (“R-1-from-R-2”) by placing a tiny electric field on R-2 that breaks the global inversion symmetry. We find that even for a tiny applied field the ensuing α_R of “R-1-from-R-2” far exceeds the effect “R-1 from trivial centrosymmetric” compound, highlighting the facts that the observed R-2 spin-splitting is not due to inadvertent breaking of the inversion symmetry in an ordinary centrosymmetric compound as recently thought¹⁹. This shows that ARPES experiments can indeed probe band splitting genuinely coming from the hidden spin-splitting, even if they are affected by surface sensitivity. This solves another criticism raised by Ref. 19 against the hidden spin polarization detection, namely the attribution of spin-splitting to surface effects rather than to the bulk. This work sheds light on the view of the recent debate around the physical meaning and relevance of the “hidden spin polarization” concept, and for the strong experimental and theoretical activity around it, motivated by the possibility to device materials with remarkable spin textures and technologically relevant properties. This work also offers clear experimental and computational frameworks to understand, tailor and utilize the R-2/D-2 effects.”

3. Referee comment: *Of course, I understand that now the authors show that the non-symmorphic symmetry is at the heart of the segregation process. But my question is: how general is this mechanism? I can imagine R-1 type Rashba systems that belong to space-groups with a symmorphic global crystal symmetry.*

Author reply: Indeed R-1 compounds do not have to maintain non-symmorphic crystal symmetries to have Rashba effect because its inversion asymmetry alone ensures the avoidance of wavefunction entanglement by lifting the degeneracy of states from the two partner sectors. This is in contrast to R-2 systems where the entanglement between the doubly degenerate states has to be prevented by some segregation-inducing symmetries (such as non-symmorphic) that suppresses the interaction between states originating from the two inversion-partner sectors and thus enables hidden spin polarization.

To clarify this as well as a few other points (see below) we have now replaced the last conclusion section of the previously submitted manuscript by an equivalent, but clearer section (p.10) called “Design principle for increasing the strength of the R-2 effect” clarifying a few points raised by the referee. In particular, comment 3 above is addressed by the yellow text below:

“Design principle for increasing the strength of the R-2 effect: R-2 materials⁶ are defined by having global inversion symmetry and two recognizable inversion-partner sectors with polar site point group symmetries (whereas layered structures make the identification of sectors easy, this applies also to non-layered materials as illustrated in Supplementary S1 for rhombohedral SnTe). Designing R-2 materials possessing large hidden spin-splitting and hence strong local spin polarization can benefit from two additional design principles:

(i) Minimizing the mixing and entanglement of the doubly degenerated wavefunctions arising from the two inversion partners sectors. Here, we point to a nontrivial mechanism of symmetry-enforced wavefunction segregation, preventing the doubly degenerate states arising from the different sectors from mixing (in contrast to the trivial physical separation of the two inversion-partner sectors). Note that R-1 compounds do not have to maintain segregation-inducing symmetries to have Rashba effect because its inversion asymmetry alone ensures the avoidance of wavefunction entanglement by lifting the degeneracy of states from the two partner sectors. The wavefunction segregation enforcing symmetry illustrated here is the non-symmorphic symmetry for bands along the $\bar{X} - \bar{M}$ direction in the BaNiS₂ BZ. Other segregation enforcing symmetry operations may exist in general cases, but they have not been discovered yet.

(ii) Instilling strong local dipole fields i.e. designing individual sectors with maximal asymmetry of the local potential within the sector. Thus, whereas the creation and enhancing Rashba effect in conventional (e.g., interfacial) Rashba materials^{2,23} entails, by tradition, breaking inversion symmetry, here our design principles for Rashba effect in centrosymmetric compounds focuses on using other symmetry operations that enhance segregation and avoid mixing. Applying the design principles (i) and (ii) one could design strong R-2 materials via selecting compounds where the wavefunctions are concentrated in real space locations that have a larger magnitude of local dipole fields. An example illustrated here is BaNiS₂. Such wavefunction segregation can be tailored through applying external electric field, strain, atom mutation, or modifications of the polar cation ordering.²³ This is illustrated by the rapid rise of α_R vs. field for bands along $\bar{X} - \bar{\Gamma}$ direction (Fig. 3a) illustrating tailoring R-2 effect. For instance, Otani and his co-workers²⁵ have recently found a strong correlation between the charge density distribution and the strength of the Rashba effect at non-magnetic metal/Bi₂O₃ interfaces. Furthermore, the unexpected rapid rise of α_R vs. field for bands along $\bar{X} - \bar{\Gamma}$ direction (Fig. 3a) implies that one might effectively tune the strength of R-2 effect. We thus present an alternative mechanism for boosting the strength of the Rashba effect, which is commonly achieved by enhancing the breaking of inversion symmetry.”

4. Referee comment: *Moreover, it seems that is a general result that R-1 and R-2 Rashba both originate from the local sector symmetries. But if I consider one of the simplest R-1 ferroelectrics, let's say alpha-Sn, how can I identify the two inversion-partner sectors such that the R-1 spin-splitting in alpha-Sn originates from the local symmetries of the sectors rather than being a consequence of the global inversion symmetry breaking?*

Author Reply: Since alpha-Sn is neither R-1 nor R-2, we guess the referee refers to alpha-SnTe, which is a R-1 ferroelectric. We have now used this excellent example to explain our unified view to R-1 materials. This is now explained in a new section in the supplementary with a new Figure S1. This section explains clearly (i) How can one identify inversion partners in non-layered materials, and (ii) how do local symmetries lead there to Rashba spin splitting effect.

“A. Identification of the two inversion-partner sectors in ferroelectric SnTe and illustration of how the R-1 spin-splitting originates from the local symmetries of the sectors rather than being a consequence of the global inversion symmetry breaking

At room temperature, α -SnTe has the centrosymmetric rocksalt (space group Fm-3m) structure¹ (Figure S1a) and, as temperature is lowered,² it undergoes a (ferroelectric) phase transition to the non-centrosymmetric (space group R3m) R-1 structure,^{3,4} where the Te atom is spontaneously displaced along the [111] direction relative to Sn atom. We focus on the bands near Fermi energy around the Z point which exhibits the most pronounced Rashba spin splitting (Figure S1b).

In the centrosymmetric rocksalt phase (illustrated by dashed lines in Figure S1b,c,d), we separate the unit cell into two inversion-partner sectors S_α and S_β (as indicated by pink and cyan in Figure S1a) connected by the Te atom, and the local dipole fields are odd around the Te inversion center (Figure S1d). However, the wavefunction is evenly distributed (guaranteed by inversion symmetry) among two inversion-partner sectors; together with the odd symmetry at the Te site. This leads according to Eq. (3) to a perfect compensation of the local dipole fields and thus vanishing Rashba effect in the centrosymmetric rocksalt phase.

However, in the non-centrosymmetric phase (illustrated by the solid lines in Figure S1b,c,d) with Te being off-center, the corresponding wavefunction becomes segregated on one of these two sectors (S_β of Figure S1e), while the local dipole fields around the displaced Te site is shifted along with the shifting of Te atom (Figure S1d). These effects give the R-1 compound α -SnTe a residual dipole field felt by band states and thus give rise to a finite Rashba spin splitting according to the unified model described by Eq. (3), similar to that of R-2 spin splitting in BaNiS₂. This illustrates the identification of the two inversion-partner sectors in non-layered SnTe and shows how the R-1 spin-splitting originates from the local symmetries of the sectors rather than being a consequence of the global inversion symmetry breaking.”

Figure S1 | Non-centrosymmetric rhombohedral α -SnTe having strong R-1 effect. Solid lines are for non-centrosymmetric R-1 case while the dashed line is for centrosymmetric case **(a)** Crystal structure of centrosymmetric alpha-SnTe identify with two inversion sectors S_α (pink domain) and S_β (cyan domain). The centrosymmetric phase transit to non-centrosymmetric rhombohedral phase with Te atom displaced from Sn along [111] direction (depict by white arrow). **(b)** Band structure along high symmetry A-Z and Z-U k-lines around the time-reversal invariant Z-point. **(c)** One-dimensional electrostatic potential (ionic + Hartree) profile along the [111] direction. **(d)** The corresponding electric dipole field. **(e)** Two-dimensional distribution of wavefunction amplitude in the (10-1) plane across Sn and Te atomic sites for the highest valence band at $k_{Z-A} = (0.509, 0.5, 0.491)$, the corresponding state is indicated by a black dot in (b).

5. Referee comment: *The study presented in this manuscript seems specific to layered systems.*

Author Reply: We do not limit our work to layered materials, however, layered materials with recognizable individual sectors are ideal candidates for uncovering the

evolution from R-2 to R-1. We stress that the conclusions drawn based on layered materials are also applicable to non-layered materials. We address this in the new supplementary section S1 (cited in full above as reply to the referee comment # 4) that describes the referee suggested example of SnTe (R3m), showing that even though it's not a layered structure, the sectors are defined and our basic Eq. (3) can be used to explain the Rashba spin splitting.

We further clarify that layered structures are not necessary by adding the sentence highlighted in blue in our reply to comment 3 above:

“(whereas layered structures make the identification of sectors easy, this applies also to non-layered materials as illustrated in Supplementary S1 for rhombohedral SnTe).”

6. Referee comment: *Some very minor observations:*

(i) In the caption of Fig. 1e, the electric field magnitude is reported to be 5 mV/Å. However in the text as well as in the panel e $E_{ext}=1$ mv/Å.

Reply: Thanks for pointing this mistake to us. We forgot to update the corresponding value in the caption and have now corrected it to 1 mV/Å.

(ii) The measure for the degree of segregation is shortened as DOD. I imagine that the authors don't want to use DOS to avoid confusion with the density of states. What the second D stand for?

Reply: Indeed, you are right. The second D stands for $D(\varphi)$. We have now replaced “DOD” by “degree of wavefunction segregation” (or “DWS”).

(iii) In the methods section, details on the first-principles calculations within an external electric field are missing.

Reply: We now added the following text to the Method section:

“We follow the approach proposed by Neugebauer and Scheffler³² to apply a uniform electric field to monolayer BaNiS₂ slab in the calculations. This approach treats the artificial periodicity of the slab by adding a planar dipole sheet in the middle of the vacuum region. The strength of the dipole is calculated self-consistently such that the electrostatic field induced dipole is compensated for.”

Comments of Reviewer #2 and authors' reply:

1. Referee comment: *The manuscript "Uncovering and tailoring hidden Rashba spin-orbit splitting in centrosymmetric crystals" explains and predicts the occurrence of the so-called "hidden spin polarization" by introducing a fundamental property of those centrosymmetric materials that sustain local sector asymmetries, i.e. the wave function "segregation" imposed by the non-symmorphic space group along certain directions in the Brillouin zone. This property has never been rationalized before in the context of materials showing "hidden spin polarization", although this is crucial to explain the emergence of this peculiar state of matter. According to the authors, the wave function segregation (i.e. wave function localization in one (a)symmetric sector) prevents the cancellation of the hidden spin splitting effects due to the equal and opposite contribution provided by the partner sector, which reestablishes the full centrosymmetry in the material.*

To itemize the main results, the paper shows that

- 1) along high-symmetry directions in the Brillouin zone, the wave function segregation is protected by a set of symmetry operations of the nonsymmorphic space group, the same set that restores the global symmetry in the compound;*
- 2) the "sector index" becomes a good quantum number (in some regions of the Brillouin zone), because the Hamiltonian is block diagonal in the "sector" representation along the same k -space directions. This disproves one of the hypotheses made in Ref. PRB 97, 125434 (2018) against the hidden spin polarization concept;*
- 3) the application of an external electric field, which automatically breaks the global inversion symmetry ($R-1$ compounds), acts in a similar way on the local sectors as the intrinsic polar field generated by local inversion asymmetries ($R-2$ materials) does. This finding unifies both global and local inversion asymmetric compounds. Despite this formal equivalence, the effect of the external local field on the band structure strongly differs according to the global or local underlying broken symmetries. However, this could be due to the breaking of the time reversal symmetry as well, see my points below;*
- 4) thanks to point 3), it is shown that ARPES experiments can indeed probe band splitting genuinely coming from the hidden spin polarization and spinorbit coupling, even if they are affected by surface sensitivity. This solves another criticism raised by Ref. 97 against the hidden spin polarization detection, namely the attribution of spin splittings to surface effects rather than to the bulk.*

The manuscript unveils the importance of the wave function segregation to explain the intriguing physical properties of the $R-2$ materials. This major property is rationalized for the first time based on symmetry arguments that constrain the form of the Hamiltonian along high-symmetry directions in the Brillouin zone. Moreover, starting from these considerations, an attempt to unify $R-1$ and $R-2$ materials is made. The paper is original, well organized. More importantly, this work is timely, in view of the recent debate around the physical meaning and relevance of the "hidden spin polarization" concept, and for the strong experimental and theoretical activity around it, motivated by

the possibility to devise materials with remarkable spin textures and technologically relevant properties.

Author reply: Thank you for appreciating the impact and scientific merits of our work.

2. Referee comment: *However, the manuscript has some flaws the authors need to correct by addressing the points below, in order of importance. The sudden increase of the α_R coefficient in the X-Gamma direction has been attributed by the authors to the wave function segregation, which kicks in along the X-Gamma direction as soon as the external electric field E_{ext} is switched on. However, the behavior of the wave function segregation content ?? is not as abrupt as the increase of α_R as a function of E_{ext} . The authors should explain why, otherwise the link between the two quantities (a key point for the paper) looks weak.*

Author reply: The “Degree of Wavefunction segregation “ (DWS) of Eqs (1)- (2) are used to describe qualitatively the trends as in Figs. 3c and 3d but the actual equation of α_R Eq. (3) with results displayed in Fig. 3a do not use DWS; indeed, as shown in Eq. (3), α_R is a sum of dipole fields weighted by corresponding wavefunction amplitudes. Clearly, the α_R does not display a simple linear correlation with the wavefunction segregation DWS because the dipole fields within a single sector are not uniform (as shown in Fig. 2e) and can even show opposite signs.

We have now added this clarification to the caption of Figure 3:

“Note that the “Degree of Wavefunction Segregation” (DWS) of Eqs. (1)-(2) are used to describe qualitatively the trends as in Figs. 3c and 3d but the actual equation of Eq. (3) with results displayed in Fig. 3a is a sum of dipole fields weighted by α_R corresponding wavefunction amplitudes and thus does not display a simple linear correlation with the wavefunction segregation DWS.”

3. Referee comment: *The authors make an attempt to unify the R-1 and R-2 class compounds by breaking the global inversion symmetry in the R-2 BaNiS₂ thanks to an external electric field. However, this breaks also the time reversal symmetry. For R-1 compounds in no-field, time reversal is generally preserved. The authors should discuss this point, in order to better justify their "R-1 from R-2" procedure.*

Author reply: We respectfully disagree on this point: As long as the magnetic field or moment is not introduced to the system, the time-reversal symmetry is preserved by electric field. We have added the following text to Page 4 Paragraph 1 to clarify it:

“..., but conserves the time reversal symmetry.”

4. Referee comment: *The wave function segregation (localization) along the M-Gamma direction has already been found and shown in Ref. 10. The localization has been quantified there by means of the integral of the local spin density operator restricted on a*

given sector. It is therefore slightly different from the D quantity defined in Eq. 2. However, the two quantities are intimately related. In Ref. 10, this finding was the basis for writing two effective decoupled Rashba Hamiltonians (one for each sector), and for estimating their effective Rashba parameters. Thus, this must be properly cited, when the authors discuss the wave function segregation in the same system and same k direction.

Author reply: Thanks for the suggestions. By saying “M-Gamma”, we assume that the referee means $\bar{X} - \bar{M}$. We have added the following clarification on bottom of Page 6:

“The DWS explicitly quantifies the locality of wavefunction, in contrast to, the implicit measure¹⁰ by means of the integral of the local spin density operator restricted on a given sector.”

Also, on page 7 first paragraph, we further clarify:
“Santos-Cottin, et al.¹⁰ had shown the localization of wavefunction in BaNiS₂ to provide the basis to decouple two effective Rashba Hamiltonians associated with each sector.”

5. Referee comment: *According to PBE+U density functional theory calculations with $U=3$ eV and $J=0.95$ eV, the lowest energy state of BaNiS₂ is indeed antiferromagnetic. However, already for $U=2$ eV the lowest energy state becomes paramagnetic. The “transition” between the antiferromagnetic and paramagnetic PBE+U ground states occurs in between 2 and 3 eV. Given the difficulty of estimating the proper value of U in the +U frameworks, it is reasonable to assume a paramagnetic state, also because the comparison with the several experimental probes strongly suggest that BaNiS₂ is a paramagnet. The most recent conductivity and susceptibility measurements of BaNiS₂ can be found in PRB 93, 125120 (2016), and could be cited.*

Author reply: Thank you for your suggestions. We have now added the following text in the second paragraph on Page 4:

“DFT+U calculations had reported that BaNiS₂ undergoes a phase transition from paramagnetic to antiferromagnetic as increasing the used U value from 2 to 3 eV. Given the difficulty of estimating the proper U value in the +U framework and experimental (conductivity and susceptibility) observation^{21,22} of metallic Pauli Paramagnet, in this work, we nevertheless adopt a non-magnetic phase for BaNiS₂ to avoid the unnecessary complications from magnetic orders.”

6. Referee comment: *Other minor points:*

(a) I found the definition of the acronym DOD misleading. It is not the abbreviation of “degree of segregation”, on the other hand DOS would be even more misleading. However, a better acronym should be found.

Reply: Thanks for pointing this out. We have now replaced “DOD” by “degree of wavefunction segregation” (DWS) and “ D ” by “ $D(\varphi)$ ”.

(b) line 330: spin quantization axis set to (0+01) direction. what is it?

Reply: (0+,0,1) is the default spin quantization axis, the notation 0+ implies an infinitesimal small positive number along x-axis. The clarification has been added to the method section on page 12:

“...the spin quantization axis is set to the default (0+,0,1) (the notation 0+ implies an infinitesimal small positive number in x-direction) with zero atomic magnetic moments.”

(c) page 12: citations above #30 are not defined, although they appear in Table I.

Reply: Sorry for the careless missing refs. Citations above #30 in Table I are now added.

(d) vaccum - vacuum

Reply: Fixed

(e) several typos (lines 69-70, line 82, line 136, line 141, line 147) in the supplementary information. Moreover, the common jargon to indicate the orbital weight of a given band, is "orbital character" and not "orbital characterization". Finally, I do not see the usefulness of Table SIV. It is never cited in the Supplementary, and reports information one can easily find in Wikipedia. Please, proofread this!

Reply: Thanks. We have corrected typos (lines 69-70, line 82, line 136, line 141, line 147) in the supplementary information. “orbital characterization” has been replaced by “orbital character”. Furthermore, we have carefully proofread the main text and the supplementary to avoid careless typos and grammatical errors.

REVIEWERS' COMMENTS:

Reviewer #1 (Remarks to the Author):

I went through the authors' reply to my concerns and found that it carefully address in a convincing way all the aspects I raised. Elaborating on the explicit example of alpha-SnTe (alpha-Sn was indeed a typo in my report) the generality and broadness of the mechanism proposed by the authors appear in a much more limpid way. It would be nice if the authors can mention GeTe besides alpha-SnTe since GeTe turns out to be a standard ferroelectric compound where a bulk Rashba effect has been predicted in 2013 [Advanced Materials 25, 509 (2013)] and recently experimentally confirmed.

I'm in favor of publication in Nature Communications.

Dr. Domenico Di Sante
Institut für Theoretische Physik und Astrophysik
University of Würzburg

Reviewer #2 (Remarks to the Author):

The authors improved their manuscript in three ways:

- 1) they clearly pointed out the novelty of their work with respect to their previous Nat. Phys. paper;
- 2) they explained why the behavior of α_R shown in Fig. 3 does not show a simple linear correlation with the wave function segregation;
- 3) they revisited the alpha-SnTe case (already published in their Nat. Phys. paper), as an R-1 example where one can identify two inversion-partner sectors where the corresponding wave function becomes segregated due to the lack of inversion symmetry in the rhombohedral phase. This is a convincing example which falls into their unified theory for R-1 and R-2 materials.

The authors replied satisfactorily to all points raised by the Referees. Therefore, I suggest the manuscript to be accepted for publication.

Typos:

in contrast to, -- in contrast to (without comma after to)

sand  and (caption of Fig. 3)

label d missing in Fig.3(d)

evenly distribute - evenly distributed (supplementary material)

Point-by-point reply to Referees' comments: COMMS-18-27301A

Reviewer #1 (Remarks to the Author):

Referee comment: I went through the authors' reply to my concerns and found that it carefully address in a convincing way all the aspects I raised. Elaborating on the explicit example of α -SnTe (alpha-Sn was indeed a typo in my report) the generality and broadness of the mechanism proposed by the authors appear in a much more limpid way.

Author Reply: Thank you for your appreciation of our work.

Referee comment: It would be nice if the authors can mention GeTe besides α -SnTe since GeTe turns out to be a standard ferroelectric compound where a bulk Rashba effect has been predicted in 2013 [Advanced Materials 25, 509 (2013)] and recently experimentally confirmed.

Author Reply: Thank you again for bring α -SnTe to us for supporting our idea. It is also a great suggestion to mention GeTe, a standard ferroelectric Rashba compound, so that we can further increase the impact of this work. We have now added following new paragraph to the main text in page 10:

“We also apply this unifying theoretical framework to a non-layered R-1 example, the α -SnTe⁶ or similarly the α -GeTe (a standard ferroelectric bulk R-1 compound predicted in 2013²⁶ and experimentally confirmed in 2016^{27, 28}), where one can identify two inversion-partner sectors and the corresponding wavefunction becomes segregated due to the lack of inversion symmetry in the rhombohedral phase (details see Supplementary Note 1). According to the unified model described by Eq. (3), such wavefunction segregation gives a residual dipole field felt by band states and thus give rise to a finite Rashba spin splitting, similar to that of R-2 spin splitting in BaNiS₂. As displacing the Te atom from Sn along [111] direction, the α -GeTe will undergo a phase transition from non-centrosymmetric rhombohedral phase to centrosymmetric rocksalt phase. We demonstrate that in the centrosymmetric rocksalt phase wavefunctions are evenly distributed among two inversion-partner sectors, leading to a perfect compensation of the local dipole fields and thus vanishing Rashba effect in the centrosymmetric rocksalt phase according to Eq. (3).”

Referee comment: I'm in favor of publication in Nature Communications.

Dr. Domenico Di Sante
Institut für Theoretische Physik und Astrophysik
University of Würzburg

Author Reply: Thank you for recommendation of publication in Nature Communications.

Reviewer #2 (Remarks to the Author):

Referee comment: The authors improved their manuscript in three ways:

- 1) they clearly pointed out the novelty of their work with respect to their previous Nat. Phys. paper;
- 2) they explained why the behavior of α_R shown in Fig. 3 does not show a simple linear correlation with the wave function segregation;
- 3) they revisited the α -SnTe case (already published in their Nat. Phys. paper), as an R-1 example where one can identify two inversion-partner sectors where the corresponding wave function becomes segregated due to the lack of inversion symmetry in the rhombohedral phase. This is a convincing example which falls into their unified theory for R-1 and R-2 materials.

Author Reply: Thank you for summarizing and confirming our improvements to this work. We have now added following new paragraph to the main text in page 10:

“We also apply this unifying theoretical framework to a non-layered R-1 example, the α -SnTe⁶ or similarly the α -GeTe (a standard ferroelectric bulk R-1 compound predicted in 2013²⁶ and experimentally confirmed in 2016^{27, 28}), where one can identify two inversion-partner sectors and the corresponding wavefunction becomes segregated due to the lack of inversion symmetry in the rhombohedral phase (details see Supplementary Note 1). According to the unified model described by Eq. (3), such wavefunction segregation gives a residual dipole field felt by band states and thus give rise to a finite Rashba spin splitting, similar to that of R-2 spin splitting in BaNiS₂. As displacing the Te atom from Sn along [111] direction, the α -GeTe will undergo a phase transition from non-centrosymmetric rhombohedral phase to centrosymmetric rocksalt phase. We demonstrate that in the centrosymmetric rocksalt phase wavefunctions are evenly distributed among two inversion-partner sectors, leading to a perfect compensation of the local dipole fields and thus vanishing Rashba effect in the centrosymmetric rocksalt phase according to Eq. (3).”

Referee comment: The authors replied satisfactorily to all points raised by the Referees. Therefore, I suggest the manuscript to be accepted for publication.

Author Reply: Thank you for appreciating the scientific merits of our work and recommend of publication.

Referee comment:

Typos:

in contrast to, -- in contrast to (without comma after to)

sand  and (caption of Fig. 3)

label d missing in Fig.3(d)

evenly distribute - evenly distributed (supplementary material)

Author Reply: Done